# Gastric Cancer Surgery: Balancing Oncological Efficacy against Postoperative Morbidity and Function Detriment

**DOI:** 10.3390/cancers16091741

**Published:** 2024-04-29

**Authors:** Andrianos Tsekrekos, Yasuhiro Okumura, Ioannis Rouvelas, Magnus Nilsson

**Affiliations:** 1Department of Clinical Science, Intervention and Technology (CLINTEC), Division of Surgery and Oncology, Karolinska Institutet, 141 52 Stockholm, Sweden; andrianos.tsekrekos@ki.se (A.T.); okumura-tky@umin.ac.jp (Y.O.); ioannis.rouvelas@ki.se (I.R.); 2Department of Surgery, University Hospital of Umeå, 907 19 Umeå, Sweden; 3Department of Upper Abdominal Diseases, Karolinska University Hospital, 141 57 Stockholm, Sweden

**Keywords:** gastric cancer, function-preserving gastrectomy, minimally invasive gastrectomy

## Abstract

**Simple Summary:**

Significant progress has been made in the surgical treatment of stomach cancer. A shift towards less invasive techniques is evident, and new methods continue to evolve. Laparoscopic surgery is widely adopted and its indications have steadily expanded, while robot-assisted surgery is being introduced in clinical practice. Function-preserving surgery, which retains the stomach’s capacity and digestive function, has now been established as a valid option for early-stage tumors. Currently, more limited procedures are being explored, such as sentinel node navigation surgery, which seeks to target only the specific lymph nodes most likely to be affected. The goal of these efforts is to reduce postoperative complications, accelerate recovery, minimize the impact on nutritional status, and preserve the quality of life of patients following surgery. This review aims to provide an up-to-date overview of the current state of surgical treatment for stomach cancer, addressing established practices, emerging trends, and future directions.

**Abstract:**

Significant progress has been made in the surgical management of gastric cancer over the years, and previous discrepancies in surgical practice between different parts of the world have gradually lessened. A transition from the earlier period of progressively more extensive surgery to the current trend of a more tailored and evidence-based approach is clear. Prophylactic resection of adjacent anatomical structures or neighboring organs and extensive lymph node dissections that were once assumed to increase the chances of long-term survival are now performed selectively. Laparoscopic gastrectomy has been widely adopted and its indications have steadily expanded, from early cancers located in the distal part of the stomach, to locally advanced tumors where total gastrectomy is required. In parallel, function-preserving surgery has also evolved and now constitutes a valid option for early gastric cancer. Pylorus-preserving and proximal gastrectomy have improved the postoperative quality of life of patients, and sentinel node navigation surgery is being explored as the next step in the process of further refining the minimally invasive concept. Moreover, innovative techniques such as indocyanine green fluorescence imaging and robot-assisted gastrectomy are being introduced in clinical practice. These technologies hold promise for enhancing surgical precision, ultimately improving the oncological and functional outcomes.

## 1. Introduction

Although medical oncology has made great advances during the last decades, including the introduction of targeted therapies and immunotherapy, which have revolutionized cancer therapy, surgery remains the backbone of the curative treatment for gastric cancer (GC). A lot has changed since the first successful gastrectomy performed by Theodor Billroth, and after more than a century of innovation and continuous evolution of the surgical technique, new methods continue to emerge and claim their place in the therapeutic arsenal.

Gastric cancer surgery has been developed with a strong main objective of maximizing the chances of cure. This objective particularly influenced the early phases of the procedure’s evolution in the direction of very radical surgery, including extensive lymphadenectomy, bursectomy, and often multiorgan resections. In the later phases of this development, a series of randomized trials were performed, allowing for an evidence-based, stepwise down-scaling of the extent of surgery, with more focus on limiting postoperative morbidity and on enhancing function preservation and quality of life. 

The present narrative review aims to summarize the state of the evidence and the steps taken leading to the current status of the surgical treatment of GC, as well as issues still under debate and future perspectives.

## 2. Standard Gastrectomy for Locally Advanced Gastric Cancer

Standard gastrectomy for locally advanced GC (LAGC) includes distal gastrectomy (DG) and total gastrectomy (TG), with the type of gastrectomy also dictating which regional nodal groups need to be dissected to eliminate any concurrent locoregional lymph node (LN) metastases (Figure 1a,b). Radicality needs to be balanced with an acceptable risk of postoperative morbidity and mortality (benefit–risk balance), and the recommendations on the specific details of the procedure to achieve this have changed over the years. One can clearly recognize a gradual transition from the early period of progressively more aggressive surgery, with prophylactic resection of adjacent anatomical structures/neighboring organs and extensive LN dissections that were thought to increase the chances of long-term survival, to the current trend of a more restrictive approach [1]. This is mainly based on the results of randomized controlled trials (RCT) conducted by the Japan Clinical Oncology Group (JCOG).

The location, growth pattern, and histological subtype of the tumor are important parameters that must be considered when it comes to determining the appropriate extent of stomach resection. According to the Japanese Gastric Cancer Treatment Guidelines and the ESMO Guidelines, a proximal margin of at least 3 cm is recommended for tumors with an expanding growth pattern (Borrmann types 1 and 2, intestinal histological subtypes) and 5 cm in cases of infiltrative growth pattern (Borrmann types 3 and 4, poorly cohesive histological subtypes) [2,3]. On the condition that an adequate proximal margin can be obtained, preservation of a small gastric remnant is preferred, as this has been shown to be beneficial in terms of postoperative quality of life [4].

### 2.1. Optimal Extent of Lymph Node Dissection and Prophylactic Splenectomy

The optimal extent of LN dissection in LAGC has been the subject of extensive research—and even intense debate—for decades. Japanese surgeons have been pioneers in mapping the lymphatic drainage of the stomach and identifying the regional LNs that are at high risk for metastasis, ultimately defining the relevant nodal groups and corresponding levels of lymphadenectomy [5]. According to the current Japanese guidelines, D2 LN dissection should be performed for tumors that are cT2 or higher and/or cN+ [2]. Although the preliminary results of Western trials were disappointing, the long-term survival benefit of D2 dissection was ultimately confirmed also in a Western setting [6,7]. D2-gastrectomy is now regarded as the standard curative LAGC procedure not only in Asia, but also in Europe, North America, and Australasia [2,8]. Attempts have been made to further improve outcomes by extending the LN dissection to include non-regional nodal groups, such as the para-aortic LNs. Although this was found to be safe when performed in high-volume centers, no survival advantage could be demonstrated [9,10,11,12]. Nevertheless, extending the dissection beyond D2 to include distinct LN stations depending on tumor location (D2+), although not yet high-level evidence-based, may be considered justified in some specific clinical situations [2].

Dissection of the splenic hilar LNs (station No. 10) was initially part of the standard lymphadenectomy when performing TG for LAGC. This was largely done by splenectomy, even though spleen-preserving techniques evolved with time [13]. The JCOG0110 trial was conducted to investigate the necessity of splenectomy for >cT2 tumors not invading the greater curvature of the stomach. This RCT found that splenectomy was associated with increased intraoperative blood loss and higher operative morbidity (pancreatic fistula and intraabdominal abscess), without conferring any survival benefit. These findings confirmed the non-inferiority of spleen preservation in terms of overall survival (OS), altering the definition of D2 LN dissection [14]. Consequently, the current guidelines suggest that splenectomy should be reserved for proximal tumors that are located on the greater curvature or directly invade the spleen [2]. However, the trial excluded patients with Borrmann type 4 GC, where splenectomy may still be justified [15,16].

### 2.2. Bursectomy and Omentectomy

Resection of the omental bursa (bursectomy) and the greater omentum (omentectomy) have traditionally been part of the procedures for advanced tumors, but their oncological value has been questioned in recent years. Bursectomy presumably removes microscopic cancer deposits that may already be present in the lesser sac, thus decreasing the risk of future relapse. However, the randomized JCOG1001 trial showed that patients who underwent bursectomy had a nearly two-fold increased incidence of pancreatic fistula without any survival benefit [17]. As a result, bursectomy is no longer performed for LAGC.

Omentectomy has also been regarded as an essential part of gastrectomy since the omentum is a common location for peritoneal recurrence. However, omentectomy has often been omitted in recent years due to the increasing use of minimally invasive surgical techniques, where omentectomy is slightly more time-consuming. Several retrospective studies comparing standard omentum resection versus preservation have shown comparable oncological outcomes, with no difference in survival and peritoneal relapse rates [18,19,20,21]. These observations have motivated the launch of two phase III RCTs (JCOG1711 and OMEGA-2) to confirm the non-inferiority of omentum preservation in LAGC [22,23].

### 2.3. Laparoscopic Surgery for Locally Advanced Gastric Cancer

Surgeons have been more reluctant to apply laparoscopic surgery for LAGC, mainly due to concerns about whether a proper D2 LN dissection could be accomplished, although early phase II trials showed that the procedure was feasible and safe [24,25]. Subsequently, three large-scale RCTs were conducted in Japan (JLSSG0901), China (CLASS-01), and South Korea (KLASS-02), comparing laparoscopic DG (LDG) to open DG (ODG) and together randomizing more than 2500 patients with stage II/III GC. The phase II part of the JLSSG0901 showed very low rates of anastomotic leakage and pancreatic fistula after LDG [26], the CLASS-01 found comparable 30-day morbidity and mortality [27], while the KLASS-02 demonstrated a lower complication rate in the LDG group [28]. Furthermore, the CLASS-01 and KLASS-02 trials showed a practically identical number of retrieved LNs between the two approaches, thus discarding doubts concerning the adequacy of LN clearance. Likewise, some skepticism has been expressed based on the concern that the capnoperitoneum and/or the laparoscopic manipulation of tumors infiltrating the serosa could aggravate the risk of peritoneal dissemination and port-site metastasis. However, an increased risk associated with LG has not been confirmed [29,30,31], and similar patterns of recurrence have been observed between LG and OG in retrospective studies [32,33]. Recently, the three RCTs reported similar 5-year OS and recurrence-free survival (RFS), confirming the non-inferiority of the laparoscopic approach [34,35,36]. Another RCT initiated in Korea (KLASS-06), comparing laparoscopic TG (LTG) and open TG (OTG) with 3-year RFS as the primary endpoint, is currently recruiting [37]. 

Although evidence on laparoscopic GC surgery originates mainly from high-incidence countries in the East, Western trials and register-based studies have also been conducted in recent years. Despite having much smaller sample sizes, Western RCTs are of importance since the study populations comprise patients subjected to neoadjuvant chemotherapy. The STOMACH trial was the first European RCT to investigate the safety and oncological quality of LTG for LAGC. All patients received chemotherapy before surgery and the majority had ≥cT3 tumors and were cN+. There was no difference between LTG and OTG with respect to the number of resected LNs and the rate of tumor-free resection margins (R0 resection), as well as the incidence of postoperative complications, including anastomotic leakage [38]. Another multicenter RCT from the Netherlands (LOGICA) including predominantly patients with LAGC (76%), of which 72% had chemotherapy before surgery, showed similar results, i.e., no difference between LG and OG regarding hospital stay, operative morbidity, R0 rates, and LN yield [39]. An Italian multicenter observational study [40] and two population-based cohort studies from the Netherlands [41] and Sweden [42] similarly showed no difference in postoperative complications between LG and OG. In addition, all three studies provide support for the oncological quality of LG, as reflected by the proportion of R0 resections and the number of retrieved LNs. 

Regarding long-term survival after LG for LAGC, only observational studies are available from Western populations. A retrospective multicenter cohort study conducted by the Italian Research Group for Gastric Cancer compared LG and OG in a propensity score-matched cohort. The authors did not find a significant difference in 3-year OS [40], in line with the results of the large Asian RCTs [34,35,36]. In a study from the United States however, based on data from the U.S. National Cancer Database, approximately 4300 minimally invasive gastrectomies (laparoscopic or robot-assisted) were matched to open procedures by means of a propensity score, and the survival analysis showed that minimally invasive surgery significantly improved OS at 5 years [43]. Most recently, the results of a population-based study with data from the Swedish national register also indicated better OS after LG, but stratified analyses revealed that the survival advantage was restricted to patients undergoing DG. On the contrary, no difference in OS was observed between LTG and OTG [42]. Nevertheless, these findings warrant cautious interpretation given the limitations inherent in the observational design.

## 3. Function-Preserving Gastrectomy for Early Gastric Cancer

The global incidence of GC demonstrates a considerable geographic variation, with the highest number of new cases recorded in East Asia. This ultimately led two high-incidence countries, Japan and South Korea, to implement population screening programs in 1983 and 2002, respectively [44,45]. As a result, approximately half of the newly diagnosed GC cases in these countries are detected at an early stage, where the prognosis is good; a 5-year disease-specific survival as high as 99% has been reported for stage IA disease (pT1a/1bN0) [46]. Consequently, given the life expectancy of these patients, preserving the health-related quality of life after surgery is essential.

Function-preserving surgery, which is indicated for early GC (EGC) and includes the pylorus-preserving gastrectomy (PPG) and proximal gastrectomy (PG), aims to preserve gastric function. The advantage of these procedures lies in minimizing the deterioration in the quality of life by reducing the occurrence of various post-gastrectomy syndromes and postoperative malnutrition, while ensuring R0 resection and adequate lymphadenectomy necessary for cure, which should always remain the main objective. Occasionally, the more limited resection necessitates the preservation of specific vessels to ensure adequate gastric blood flow, making lymph node (LN) dissection more challenging. Function-preserving gastrectomy is suitable for EGC since (1) the frequency of LN metastasis is low and usually limited to first tier lymph node stations, meaning that a D2 lymphadenectomy is not necessary and a more limited dissection (D1 or D1+) sufficient, and (2) a macroscopic resection margin of 2 cm is considered adequate [2]. 

### 3.1. Laparoscopic Surgery for Early Gastric Cancer

Gastrectomy for EGC, including PPG and PG, is increasingly performed using minimally invasive surgical modalities. Shortly after the first report on LDG in the 1990s, small-scale trials confirmed the perceived short-term benefits of the less invasive nature of the procedure (reduction in intraoperative bleeding and postoperative pain, less impairment of pulmonary function, shorter hospitalization) [47,48,49]. Larger studies followed and demonstrated decreased morbidity and improved quality of life after LDG, while the oncological outcomes were not inferior compared to ODG [50,51,52,53,54]. Ultimately, two RCTs from Korea (KLASS-01) and Japan (JCOG0912) published their long-term results in 2019, showing no differences between LDG and ODG with regard to 5-year OS and RFS, respectively [55,56]. The next logical step was the application of the laparoscopic technique in TG, with early retrospective studies indicating an increased rate of anastomotic leakage [57,58]. However, this was not confirmed in two prospective single-arm trials (JCOG1401 and KLASS-03), in which laparoscopic construction of the esophagojejunostomy was found to be safe [59,60]. The CLASS02 trial, which randomized patients with stage I GC between LTG and OTG, came to the same conclusion; overall morbidity and mortality rates, including the occurrence of anastomotic leakage, did not differ between the two groups [61]. A meta-analysis including 19 studies—with subgroup analyses for EGC and LAGC—has shown comparable 5-year OS after LTG and OTG, but data on long-term survival are otherwise relatively limited [62].

### 3.2. Pylorus-Preserving Gastrectomy

Pylorus-preserving gastrectomy can be performed for EGC located in the middle third of the stomach (Figure 2a) [2]. As indicated by the name of this procedure, the difference between PPG and conventional DG is the preservation of the pylorus, which functions as the physiologic regulator of gastric emptying. Several technical details are crucial for a good functional outcome. Most importantly, both the blood supply and the innervation of the pyloric sphincter need to be preserved [63]. The blood supply of the pylorus comes mainly from the infrapyloric artery [64], while its innervation comes from the hepatic branch of the anterior vagal trunk. The pyloric nerve branches run along the right gastric artery and, since they are to be preserved, dissection of the corresponding LNs is omitted. Several studies have shown that in cases of EGC in the middle third of the stomach, the LNs along the proximal part of the right gastric artery, including its first branch (LN station No. 5), and the proximal part of the right gastroepiploic artery (LN station No. 6) are rarely affected by metastases [65,66,67]. Hence, it is deemed oncologically justified to refrain from dissection of LN station No. 5 when performing PPG. Although no precise criterion dictates the distance from the pylorus ring at which the stomach should be transected, a common practice is to spare approximately 4 cm of the antrum; this has been shown to reduce the risk of early gastric stasis, a common problem that was encountered postoperatively when this procedure was first introduced. The proximal side of the stomach is then divided, ensuring an adequate distance from the upper border of the tumor. While conventional practice previously involved hand-sewn anastomosis, contemporary reports have highlighted the feasibility of different methods of totally laparoscopic anastomosis utilizing linear staplers [68,69].

The oncological safety of PPG has been demonstrated in a large multicenter cohort study from Japan, which showed no significant difference in long-term survival compared to DG [70]. Reported advantages of PPG encompass less postoperative body weight loss and a reduction in the occurrence of postoperative dumping syndrome and anemia. Apart from a well-maintained nutritional status, PPG is also associated with only mild fluctuations of postprandial blood glucose levels [71,72,73]. Although the Korean KLASS-04 trial revealed comparable 30-day morbidity between laparoscopic PPG and LDG [74], drawbacks of the PPG exist and include the increased risk for gastric stasis and reflux esophagitis. The reported incidence of postoperative gastric stasis after PPG is in the range of 6–8% [74,75]. While the primary etiology is attributed to the disruption of nerves responsible for gastric peristalsis, Takahashi et al. also identified advanced age, diabetes mellitus, and intraabdominal infection after surgery as risk factors for gastric stasis [75]. However, in a study evaluating the long-term functional results following PPG, Nunobe et al. did not find any significant differences compared to DG with Billroth I reconstruction [76]. In addition, in a meta-analysis including over 1200 patients, Xiao et al. confirmed the aforementioned functional advantages of PPG and showed that, although gastric stasis was more common, the symptoms subsided over time [77]. Postoperative reflux esophagitis is another clinical concern after PPG. Otake et al. reported an incidence of approximately 15% (endoscopically verified, Los Angeles grade B or higher), with male gender, preoperative reflux esophagitis, high body mass index, presence of hiatal hernia, and prolonged gastric stasis after surgery being significant risk factors for the development of this mid- to long-term complication [78].

### 3.3. Proximal Gastrectomy

Proximal gastrectomy is indicated for EGC in the upper third of the stomach, provided that at least half of the distal stomach can be preserved (Figure 2b) [2]. The alleged advantages of this approach are believed to be greater when preservation of two-thirds of the stomach is feasible [79]. In PG, no LN dissection is performed along the right gastric artery (LN stations No. 5 and No. 3b) and the right gastroepiploic artery (LN stations No.6 and No.4d), with these vessels preserved. This is justified by the low incidence of metastatic involvement of these LNs in EGC located in the proximal stomach [80]. 

Long-term prognosis is reported to be equivalent after PG compared to TG for proximally located EGC [81]. At the same time, PG has demonstrated superior nutritional outcomes, as reflected by the lower percentage of body weight loss and the decreased requirement for vitamin B12 supplementation after surgery [81,82,83,84]. The advantage of PG is attributed to the preservation of the physiological functions (mechanical and chemical digestion, secretion of intrinsic factor, etc.) and reservoir functions of the stomach. However, an increased risk of postoperative reflux esophagitis has been reported after PG [85]. Yamasaki et al. found a significantly higher occurrence of esophagitis after PG compared to TG, despite otherwise comparable overall complication rates [81]. Given the persisting acid secretion in the preserved gastric body, preempting postoperative reflux esophagitis is of great importance, and consequently, reconstruction techniques that simultaneously address this problem are necessary. 

Various reconstruction methods have been described following PG and can be broadly classified into two categories: those entailing anastomosis of the remnant stomach to the esophagus and those employing the jejunum for reconstruction. Esophagogastric anastomotic techniques with integrated anti-reflux properties have been developed, such as the double flap technique (DFT) and the side overlap with fundoplication by Yamashita (SOFY). In DFT, a seromuscular stomach wall flap is created and subsequently wrapped around the esophagogastrostomy to prevent reflux [86,87]. In a multicenter retrospective study with 464 patients, the incidence of esophagitis grade B or higher one year after DFT was 6% [88]. Conversely, SOFY involves anastomosing the esophagus and remnant stomach using a linear stapler, with fixation of the esophagus to the stomach at a 90-degree angle to avert reflux [89]. Anastomotic methods utilizing the jejunum encompass the jejunal interposition and the double tract reconstruction (DTR) [90,91]. The DTR, being more suitable for a laparoscopic approach, has gained precedence over jejunal interposition. A recent meta-analysis of studies comparing DFT and DTR found that DFT is more time-consuming but has an advantage over DTR in terms of better postoperative nutritional status [92]. Nonetheless, high-grade evidence is still lacking, and a consensus on the optimal reconstruction method after PG remains to be determined.

## 4. Cancer of the Esophagogastric Junction

One of the issues still under debate concerns the management of tumors affecting the esophagogastric junction (EGJ), both in terms of the type of resection and in terms of the appropriate extent of LN dissection. Several factors contribute to the current differences in the surgical approach when it comes to tumors arising in the vicinity of the EGJ: (1) there is no global consensus regarding the classification of EGJ-cancers, with the Siewert classification used in the West [93] and the Nishi classification in the East [94]; (2) it is not uncommon that adenocarcinomas and squamous cell carcinomas of the EGJ are lumped together in clinical trials; and (3) both thoracic surgeons and gastrointestinal surgeons are involved in the treatment of these patients, and practices vary considerably across countries and institutions. In an attempt to promote harmonization in clinical practice, the Japanese Gastric Cancer Association and the Japan Esophageal Society conducted a prospective multicenter study and defined the LN stations that need to be dissected in cT2-4 EGJ-tumors, depending on the length of esophageal involvement [95]. In addition, an international consensus meeting was held in 2023 to establish international clinical practice guidelines, and the recommendations resulting from this summit were recently published [96]. Following a comprehensive review of the existing evidence, including several meta-analyses, the authors concluded that in cases of EGJ cancer, the lower mediastinal and suprapancreatic LN stations should be dissected. However, utilization of left thoracic access was discouraged due to the increased morbidity and lack of benefit in terms of long-term survival, based on the results of the JCOG9502 trial [97,98] as well as the results of a meta-analysis including nine retrospective studies [96]. Differences are also apparent with respect to the chosen type of resection. In the West, it is common practice to regard Siewert type III EGJ tumors as GC and treat them with TG, while types I and II are usually subjected to esophagectomy. Nevertheless, the optimal approach for type II tumors is still to be determined and is the subject of an ongoing RCT (CARDIA-trial) [99]. In the East, on the other hand, based on the paucity of metastatic involvement of LN stations No. 4d, 5, and 6 in EGJ cancer [95], PG encompassing lower esophagectomy is also considered a valid option. However, LN stations No. 19, 20, and 110 must be included [3,100]. Minimally invasive surgical techniques can be applied in cases where a transthoracic approach is indicated, and robot-assisted surgery, in particular, is associated with a significantly lower incidence of postoperative complications and better QoL, without compromising long-term survival [96]. 

## 5. Emerging Techniques in Gastric Cancer Surgery

### 5.1. Indocyanine Green Fluorescence Imaging-Guided Lymphadenectomy and Sentinel Node Navigation

Indocyanine green (ICG) near-infrared fluorescent imaging is a relatively new technology with several potential clinical applications [101]. This modality is now integrated into most modern devices used in laparoscopic and robot-assisted surgery (Figure 3). In the setting of GC surgery, ICG-guided imaging is mainly used to assist in LN dissection by individually mapping the perigastric lymphatic channels and lymph nodes with high risk of involvement. In a recent Chinese trial, patients undergoing LG were randomized between standard surgery and LG utilizing ICG-guided imaging. The study showed that the use of ICG resulted in a significantly higher number of harvested LNs, without increasing the operating time [102]. This trial recently reported interesting long-term results, showing significantly better 3-year OS and DFS for the ICG group [103]. 

Although a high LN yield is an indicator of good surgical quality, the main contribution of ICG imaging seems to be in the opposite direction; identification of the first lymph node tier for the individual primary tumor location—in effect the sentinel node(s) (SN)—during surgery for EGC and confirmation that no metastases are present may ultimately allow for local gastric resections with much more limited lymphadenectomy (“SN basin dissection”). Sentinel node navigation surgery (SNNS) for GC has been studied for more than 20 years, and the concept is similar to the one already applied in melanoma and breast cancer surgery, although GC has a considerably more complex and variable lymphatic drainage and skip metastases may occur [104]. A Korean multicenter RCT (SENORITA) compared standard LG to laparoscopic SNNS and demonstrated that the 5-year survival outcomes were similar between the two groups [105]. Another prospective, non-randomized phase III study is underway in Japan [106]. An additional crucial issue is whether the SNNS concept is valid following non-curative endoscopic submucosal dissection (ESD), given that the lymphatic flow from the previous tumor area could be altered. The results of a Japanese retrospective study suggest that SNNS is still feasible in this clinical scenario [107], and the ongoing SENORITA 2 trial aims to definitely answer this question [108]. The ultimate goal is to further modify and individualize gastric surgery to minimize the extent of resection without compromising the oncological outcome.

### 5.2. Robot-Assisted Laparoscopic Gastrectomy

Similar to many other fields in abdominal and pelvic surgery, robot-assisted surgery for GC is already implemented in clinical practice in many institutions worldwide. The premise is that robot-assisted gastrectomy (RAG) offers the same benefits as LG, while addressing many of the ergonomic disadvantages of conventional laparoscopic surgery. This technology provides high-resolution 3-dimensional imaging through a stable camera controlled by the surgeon, as well as tremor elimination and increased degrees of freedom by the use of articulated instruments, together contributing to a superior operative environment. Nevertheless, the conceivable advantage of enhanced precision comes with a considerably higher cost per procedure, which has been the major argument against adopting this technology. Again, Asian surgeons have been pioneers in exploring the feasibility, safety, and efficacy of RAG. The results of the first two RCTs comparing RAG to LG were published in 2021; both trials showed that RAG was associated with reduced postoperative morbidity and faster recovery. In addition, RAG resulted in a higher number of resected LNs in one of the trials [109,110]. These findings were later confirmed on a large, propensity score-matched cohort including more than 3500 patients [111]. However, the longer operation time and higher costs are still a concern. A Japanese phase III RCT is ongoing, aiming to confirm the superiority of RAG over LG in terms of postoperative morbidity (JCOG1907, MONA LISA study) [112]. Different techniques for resection, reconstruction, anastomosis, and lymphadenectomy are being explored by surgeons and are constantly refined [113]. The results will most probably improve as more experience is acquired, but whether RAG will ultimately benefit patients in terms of better long-term survival is yet to be proven.

## 6. Conclusions and Future Directions

Significant progress has been made in the surgical management of GC over the years. In parallel to improvements in the surgical technique, advances in anesthesiology, prehabilitation, and nutritional support, as well as the implementation of enhanced recovery after surgery (ERAS) protocols have contributed to better outcomes [114]. Since the first report in the early 1990s, LG has been widely adopted and its indications have gradually expanded, from EGC located at the distal part of the stomach, to LAGC where TG is required. Thus, conventional open surgery is now reserved for LAGC where a complex, multivisceral resection is needed. On the other hand, function-preserving LG has also emerged as a valid option for EGC, improving the postoperative quality of life of the patients. These advancements underline the importance of constantly refining surgical techniques to optimize patient outcomes. The process of implementing minimally invasive GC surgery has been accelerated by the fact that recent studies have challenged traditional practices, questioning the necessity of procedures such as splenectomy, bursectomy, and omentectomy in light of their impact on operative morbidity without any clear survival benefits. Increased awareness of the importance of benefit–risk balance lies behind the current trend of less extensive surgery, leading today’s surgeons to go even further and explore the role of SNNS in EGC. If methods that accurately exclude LN metastases intraoperatively are developed and prove to be reliable, then even more limited, individualized procedures for patients with EGC could become a reality. 

Looking ahead, future perspectives in GC surgery center on innovative techniques such as ICG fluorescence imaging-guided lymphadenectomy and RAG. Many centers worldwide have already taken the next step, introducing these modalities in clinical practice. These technologies hold promise for enhancing surgical precision, potentially improving the oncological outcomes. However, the increasing costs related to the equipment used in modern surgery are a concern for healthcare providers and policy-makers. Indeed, in the era of value-based medicine, investment into more expensive technologies can be justified only if significant long-term benefits are apparent and reflected in patient outcomes. However, the increased operative costs could be counterbalanced by, for example, decreased postoperative complications, shorter hospitalization, and shorter sick leave. The cost–benefit performance of RAG is an issue that cannot be overlooked and warrants investigation.

The results of clinical research originating from high-incidence countries in the East have a big impact on the Western hemisphere. Consequently, previous differences in surgical practice have gradually been lessened. Nevertheless, it should be kept in mind that the favorable results following minimally invasive GC surgery, as well as the promising results of other novel technologies, are reported by experienced surgeons in high-volume centers. Thus, in low-incidence countries in the West, centralization of GC surgery is necessary to achieve similar results. Additionally, this is also a prerequisite for high-quality clinical research to be possible. Overall, the continuous advancements in surgical management offer hope for further improvement of the standard of care for patients diagnosed with GC.

## Figures and Tables

**Figure 1 cancers-16-01741-f001:**
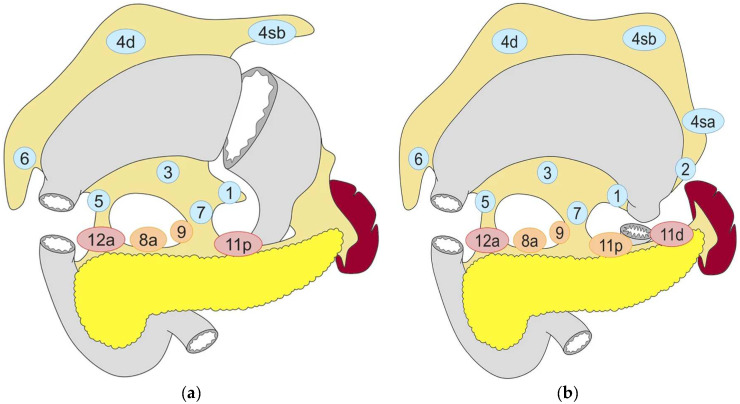
Lymph node dissection in (**a**) distal gastrectomy and (**b**) total gastrectomy. Lymph node stations in blue need to be dissected in D1 dissection. In addition, lymph node stations in orange need to be dissected in D1+ dissection and lymph node stations in red need to be dissected in D2 dissection. Reproduced under the Creative Commons CC BY Attribution 4.0 International License from reference [2].

**Figure 2 cancers-16-01741-f002:**
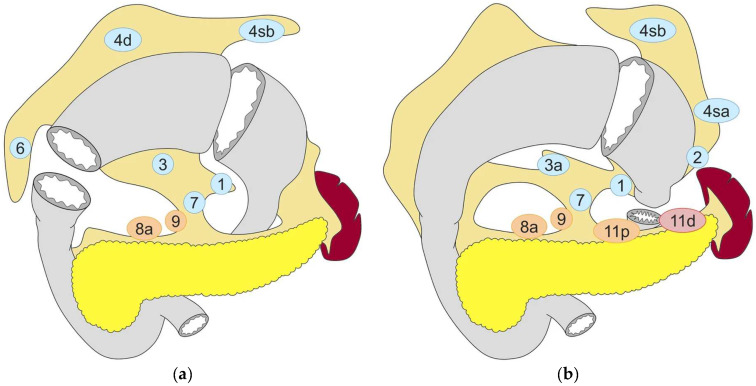
Lymph node dissection in (**a**) pylorus-preserving gastrectomy and (**b**) proximal gastrectomy. Lymph node stations in blue need to be dissected in D1 dissection. In addition, lymph node stations in orange need to be dissected in D1+ dissection and lymph node stations need to be dissected in D2 dissection. Reproduced under the Creative Commons CC BY Attribution 4.0 International License from reference [2].

**Figure 3 cancers-16-01741-f003:**
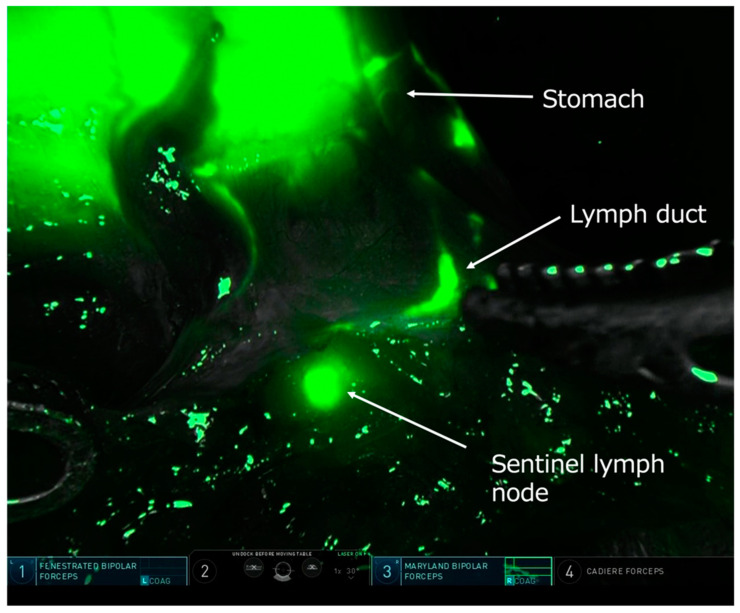
Intraoperative near-infrared fluorescent light image during robot-assisted gastrectomy showing sentinel lymph node mapping after injection of indocyanine green around the tumor. A lymphatic vessel and a sentinel node are clearly visualized. Courtesy of Dr. Koshi Kumagai, Kitasato University, Sagamihara, Japan.

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
