# Peer review of "Gastric Cancer Surgery: Balancing Oncological Efficacy against Postoperative Morbidity and Function Detriment"

_cancers, 2024, doi:10.3390/cancers16091741_

Round 1
Reviewer 1 Report
Comments and Suggestions for Authors
The researchers of the present work offer us the opportunity to evaluate the various stages of gastric surgery between history and contemporaneity from the distant times of simple gastric resection to today. They make no mention of diagnostics and possible neo/adjuvant therapy. Today, from a purely surgical point of view, we are in fact experiencing the possibility of being able to choose, based on the patient's pathology, the most appropriate treatment between open, laparoscopic and robotic surgery with lymphadenectomies suited to the patient's pathology up to D2 plus (doi.org /10.3390/cancers16071376 to be cited in the bibliography) which offers the patient, together with chemotherapy treatment, a good life perspective. The paper also takes into consideration the quality of life and the reconstruction methods that have the least impact on the nutritional status to guarantee a return to the pre-operative performance status in the shortest possible time. In our clinic in the late 90s (PMID: 9973791) we managed to obtain a good performance status in 18 months. But currently with preoperative immunotherapy and ERAS treatment (DOI: 10.1007/s12032-018-1153-0) we can achieve excellent results more quickly. Excellent work with bibliography that supports the initial theses, in good English, with substantial bibliography
Comments on the Quality of English Languagegood english
Author Response
Question/Comments: “The researchers of the present work offer us the opportunity to evaluate the various stages of gastric surgery between history and contemporaneity from the distant times of simple gastric resection to today. They make no mention of diagnostics and possible neo/adjuvant therapy. Today, from a purely surgical point of view, we are in fact experiencing the possibility of being able to choose, based on the patient's pathology, the most appropriate treatment between open, laparoscopic and robotic surgery with lymphadenectomies suited to the patient's pathology up to D2 plus (doi.org /10.3390/cancers16071376 to be cited in the bibliography) which offers the patient, together with chemotherapy treatment, a good life perspective. The paper also takes into consideration the quality of life and the reconstruction methods that have the least impact on the nutritional status to guarantee a return to the pre-operative performance status in the shortest possible time. In our clinic in the late 90s (PMID: 9973791) we managed to obtain a good performance status in 18 months. But currently with preoperative immunotherapy and ERAS treatment (DOI: 10.1007/s12032-018-1153-0) we can achieve excellent results more quickly. Excellent work with bibliography that supports the initial theses, in good English, with substantial bibliography.
Answer: We thank the reviewer for the positive feedback and recommendations on additional references for the bibliography. We have reviewed 2 of the 3 articles proposed (we were not able to review one article as it is written in Italian) and agree that ERAS should be mentioned in our paper. We have therefore added another reference and the following sentence in the Conclusion and Future Directions section:
“In parallel to improvements in the surgical technique, also advances in anesthesiology, prehabilitation, and nutritional support, as well as the implementation of enhanced recovery after surgery (ERAS) protocols have contributed to better outcomes [114].”
Reference added:
- Mortensen, K.; Nilsson, M.; Slim, K.; Schäfer, M.; Mariette, C.; Braga, M.; Carli, F.; Demartines, N.; Griffin, S. M.; Lassen, K., Consensus guidelines for enhanced recovery after gastrectomy: Enhanced Recovery After Surgery (ERAS®) Society recommendations. The British journal of surgery 2014, 101 (10), 1209-29.
Reviewer 2 Report
Comments and Suggestions for Authors
Thank you for permitting me to review this manuscript
In this narrative review paper , the authors are updating current knowledge on surgical treatement of gastric cancer
Please develop this phrase and add some references (paragraph 2)
The location, growth pattern and histological subtype of the tumor are important parameters that must be considered when it comes to determining the appropriate extent of stomach resection
Indocyanin imaging
If possible please provide a picture with images and explanations in order to facilitate memorisation of this technique for the reader
. Cancer of the esophagogastric junction
Please add in a 1 phrase or 2 , the summary of the new international recommendation
End of discussion
Please detail the postoperative morbidity somewhere in the discussion or in the introduction
Author Response
Question/Comment: Please develop this phrase and add some references (paragraph 2)
“The location, growth pattern and histological subtype of the tumor are important parameters that must be considered when it comes to determining the appropriate extent of stomach resection”.
Answer: We believe that this phrase is adequately explained in the sentence that follows, where we refer to the ESMO and the JGCA guidelines and the recommended minimal surgical margin depending on macroscopic (Borrmann) type and histology (references 2 and 3):
“According to the ESMO Guidelines and the Japanese Gastric Cancer Treatment Guidelines, a proximal margin of at least 3 cm is recommended for tumors with an expanding growth pattern (Borrmann types 1 and 2, intestinal histological subtypes) and 5 cm in cases of infiltrative growth pattern (Borrmann types 3 and 4, poorly cohesive histological subtypes) [2, 3]. On the condition that an adequate proximal margin can be obtained, preservation of a small gastric remnant is preferred, as this has been shown to be beneficial in terms of postoperative quality of life [4].”
Question/Comment: Indocyanine imaging
If possible, please provide a picture with images and explanations in order to facilitate memorisation of this technique for the reader.
Answer: We have added a figure showing an intraoperative ICG image (Figure 3).
Question/Comment: Cancer of the esophagogastric junction
Please add in a 1 phrase or 2, the summary of the new international recommendation
Answer: Concerning the recommendations of the Upper GI Oncology Summit 2023 (reference [96]), we believe that relevant issues are already covered by the following sentences in our manuscript:
“In addition, an international consensus meeting was held in 2023 to establish international clinical practice guidelines, and the recommendations resulting from this summit were recently published [96]. Following a comprehensive review of the existing evidence, including several meta-analyses, the authors concluded that in cases of EGJ cancer, the lower mediastinal and suprapancreatic LN stations should be dissected. However, utilization of left thoracic access was discouraged, due to the increased morbidity and lack of benefit in terms of long-term survival, based on the results of the JCOG9502 trial [97, 98] as well as the results of a meta-analysis including 9 retrospective studies [96].” ... “Minimally invasive surgical techniques can be applied in cases where a transthoracic approach is indicated, and robot-assisted surgery, in particular, is associated with a significantly lower incidence of postoperative complications and better QoL, without compromising long-term survival [96].”
Question/Comment: End of discussion
Please detail the postoperative morbidity somewhere in the discussion or in the introduction.
Answer: Since many different surgical methods are presented, we believe it will be easier for the reader if morbidity data is presented separately in each section of the paper. We have therefore decided to present postoperative morbidity by procedure type, please see:
Page 4, section 2.3, 1st paragraph
Page 4, section 2.3, 2nd paragraph
Page 5, section 3.1
Page 6-7, section 3.2, 2nd paragraph
Page 7, section 3.3, 2nd paragraph
Page 9, section 5.2
Reviewer 3 Report
Comments and Suggestions for Authors
The Authors present an extremely well written and comprehensive review on surigical management of gastric cancer. I feel that this paper has a high chance of becoming a reference in the field.
Only one comment: can the Authors briefly comment on the criteria applied for literature research?
Comments on the Quality of English LanguageFine, but of course proof-reading will be necessary before publication.
Author Response
Question/Comment: Can the Authors briefly comment on the criteria applied for literature research?
Answer: We thank the reviewer for the positive feedback. The current review is narrative in nature, not a systematic review, meaning that a systematic literature search was not conducted. We now clarify that in the Introduction section.